# Data-driven prior elicitation for Bayes factors in Cox regression for nine subfields in biomedicine

**Maximilian Linde**[1,2]*, **Laura Jochim**[3], **Jorge N. Tendeiro**[4], **Don van Ravenzwaaij**[2]

**1** GESIS - Leibniz Institute for the Social Sciences, Cologne, Germany, **2** Department of Psychometrics and Statistics, University of Groningen, Groningen, The Netherlands, **3** Utrecht University, Utrecht, The Netherlands, **4** Graduate School of Advanced Science and Engineering, Hiroshima University, Hiroshima, Japan

* maximilian.linde@gesis.org

**Data availability statement:** Data used in section 'Example Application' was shared with the authors by NIAID and may not be distributed

## Abstract

Biomedical research often utilizes Cox regression for the analysis of time-to-event data. The pervasive use of frequentist inference for these analyses implicates that the evidence for or against the presence (or absence) of an effect cannot be directly compared and that researchers must adhere to a predefined sampling plan. As an alternative, the use of Bayes factors improves upon these limitations, which is especially important for costly and time-consuming biomedical studies. However, Bayes factors involve their own difficulty of specifying priors for the parameters of the statistical model. In this article, we develop data-driven priors centered around zero for Cox regression tailored to nine subfields in biomedicine. To this end, we extracted hazard ratios and associated $x$% confidence intervals from the abstracts of large corpora of already existing studies within the nine biomedical subfields. We used these extracted data to inform priors for the nine subfields. All of our suggested priors are Normal distributions with means of 0 and standard deviations closely scattered around 1. We propose that researchers use these priors as reasonable starting points for their analyses.

## Introduction

The collection and analysis of time-to-event data forms a central part of modern biomedical research (see, e.g., [1–6]). In these types of designs, the outcome variable is the time until an event of interest occurs, which is commonly called the survival time. In the medical context, this event of interest could be, for example, death, relapse towards alcoholism, disease/symptom onset, or recovery. Using survival analysis (we refer the interested reader to other sources, which treat survival analysis more thoroughly; e.g., [7–15]), it is possible to estimate differences in event rates between conditions, which makes it particularly appealing for clinical trials. For instance, evidence about the effectiveness of an oncological treatment of cancer patients could be gathered by comparing the survival times of patients receiving the treatment to the survival times of patients receiving a placebo or an active control treatment (for examples see [1,16,17]).

beyond the first and last author (for more information, see
https://accessclinicaldata.niaid.nih.gov/). All other data may be obtained from the OSF page associated with the project:
https://osf.io/ua4ys/.

**Funding:** This research was supported by a Dutch scientific organization VIDI fellowship grant (016.Vidi.188.001) awarded to Don van Ravenzwaaij and a Japanese JSPS KAKENHI grant (21K20211) awarded to Jorge N. Tendeiro. Neither of the funders played any role in the study design, data collection and analysis, decision to publish, or preparation of the manuscript.

**Competing interests:** The authors have declared that no competing interests exist.

The use of frequentist inference for the analysis of survival data has a long tradition in biomedical research and is still very common today [18]. Classical frequentist inference, however, is not well suited to quantify evidence in favor of the absence of an effect. In addition, frequentist inference requires fixing the study sample size in advance to avoid an inflation of the Type I and/or Type II error rates [19], a downside that can be particularly costly in the realm of resource-intensive biomedical research and clinical trials [18,20,21]. A prominent way to deal with this issue within the frequentist framework is the use of group sequential designs, whereby sequential testing and optional stopping of data collection is permitted at the cost of implementing some procedure that controls the Type I and/or Type II error rates. These designs have a long and successful tradition in biomedicine (e.g., [22]). As an alternative, Bayesian statistics have gained popularity among researchers [23]. In particular, Bayes factors [24–27] do not suffer from the two limitations mentioned before and are therefore a valuable alternative for conducting inference in biomedical research [28].

Bayesian modeling with the goal of either hypothesis testing or parameter estimation requires the specification of a prior distribution for the parameters of the model. The prior expresses one's beliefs about the plausibility of parameter values before observing the data [29–31]. Oftentimes, it is notoriously difficult to express these beliefs. Moreover, within Bayesian hypothesis testing, different priors sometimes lead to qualitatively different Bayes factors [30,32–36]. As a result, some researchers lament that the use of Bayesian statistics involves subjectivity (e.g., [37]) and that proper guidance is missing, possibly resulting in hesitation to use Bayesian inference. Hence, recommendations for well-established default priors in Bayesian survival analysis - in particular Cox proportional hazards regression (henceforth called Cox regression or Cox model; [38]) - are missing and urgently needed.

In this article, we propose default priors for Bayesian Cox regression tailored to nine subfields within biomedicine. The construction of these priors harnesses historical records consisting of large corpora of hazard ratios and associated $x$% confidence intervals from existing studies within the respective subfields. We argue that these proposed priors can be used as reasonable defaults or starting points for biomedical researchers wishing to conduct a Bayesian Cox regression.

The remainder of this article is structured as follows: First, we provide an overview of Cox regression and Bayes factors. Second, we briefly review how priors can be defined. Third, we explain our process of generating priors for nine subfields in biomedicine and present the corresponding results. Fourth, we reflect on our findings and implications thereof, and conclude with recommendations.

## Bayes factors in Cox regression

Survival analysis is a statistical method to analyze time-to-event/survival data. Among the many existing forms of survival analysis - for example, Kaplan-Meier product-limit estimator [39], parametric survival analyses (e.g., Exponential, Gompertz, and Weibull), and Cox regression [38] - the latter is used most frequently within biomedicine (e.g., [10]). Therefore, our treatment of priors for survival analysis is limited to the case of Cox regression.

In Cox regression, the hazard function $\lambda(t)$ presents the risk of an event happening in a small time period around a specific time $t$ within cases for which the event has not yet happened before time $t$ (see, e.g., [7,8]). The specific $\lambda(t)$ is allowed to have any shape but must be proportional across all values of an independent variable $x$. Usually, the main goal is not to estimate $\lambda(t)$ but rather to estimate the $\beta$ parameter of the Cox model:

$$\lambda(t \mid x) = \lambda(t) e^{x\beta}. \tag{1}$$

Here and throughout, we work with the specific case where $x$ is dichotomous and dummy-coded (i.e., there are two conditions, a common situation in biomedical designs). For this scenario, a hazard ratio can be calculated

$$\text{HR} = e^{\beta}, \tag{2}$$

which provides information about the relative hazard rates between conditions.

In clinical trials, HR is often the key indicator regarding the effectiveness of a treatment. HR = 1 (or $\beta$ = 0) means that the two conditions have the same risk of the event happening at any $t$; HR > 1 (or $\beta$ > 0) means that the experimental condition has a higher risk of the event happening at any $t$; HR < 1 (or $\beta$ < 0) means that the control condition has a higher risk of the event happening at any $t$. Frequentist inference on HR or $\beta$ is then conducted either in the form of null hypothesis significance testing (i.e., test statistic and $p$-value) or in the form of estimation (i.e., a point estimate accompanied with a confidence interval).

The reliance on frequentist inference [40,41] has some undesirable consequences for biomedical research. Here, we focus on two of these consequences, namely (1) the impossibility to obtain evidence for the null hypothesis and (2) the inability to adjust the sampling plan based on interim results. Concerning (1), it is important to not only determine whether a treatment is working but also whether a treatment is *not* working over and above a placebo effect. The frequentist approach is not suitable for this because it only allows rejecting the hypothesis that there is no effect, but not accepting it (e.g., [35,42–46]).

Concerning (2), the use of frequentist inference prescribes the diligent adherence to a pre-defined sampling plan, prohibiting to continue or prematurely stop data collection based on interim data analyses (e.g., [47–50]). Further criticism is described elsewhere (see, e.g., [41,46,51–53]).

Bayesian testing in the form of Bayes factors permits a direct comparison between the evidence for the null hypothesis that there is no effect and an alternative hypothesis that operationalizes that there is some effect [42]. For instance, with $\text{BF}_{10}$ = 14, it allows the interpretation that the obtained data is 14 times more likely under the chosen hypothesis that there is some effect compared to the hypothesis that there is no effect; similarly, $\text{BF}_{10}$ = 0.2 indicates that the obtained data is 1/0.2 = 5 times more likely under the hypothesis that there is no effect compared to the chosen hypothesis that there is some effect.

Moreover, using Bayes factors, it is legitimate to monitor the results and stop data collection once a predetermined evidence threshold is reached [47–50,54,55]. Thus, Bayes factors take the evidence for and against both the null and alternative hypotheses into account, yield more substantial interpretations, and empower researchers to sample just the sufficient amount of cases. These characteristics of Bayes factors are critical for biomedical research as studies (especially clinical trials) can be expensive and time-consuming.

In the case where there is a point null hypothesis stating that there is no effect

$$\mathcal{H}_0 : \beta = 0 \tag{3}$$

and an interval alternative hypothesis stating that there is an effect

$$\mathcal{H}_1 : \beta \sim f(\phi), \tag{4}$$

the Bayes factor is a ratio of a marginal likelihood and a likelihood evaluated at $\beta$ = 0. Here, $f(.)$ represents any probability density function and $\phi$ the associated parameters. Let $\Omega_1$ be

the parameter space under the alternative hypothesis; then the Bayes factor is:

$$\mathrm{BF}_{10} = \frac{P(D \mid \mathcal{H}_1)}{P(D \mid \mathcal{H}_0)} = \frac{\int_{\beta \in \Omega_1} \overbrace{f(D \mid \beta)}^{\text{Likelihood}} \overbrace{f(\beta)}^{\text{Prior}} \mathrm{d}\beta}{\underbrace{f(D \mid \beta = 0)}_{\text{Likelihood at } \beta = 0}}. \tag{5}$$

In Eq (5), the integral constitutes a weighted average of the likelihood, with weights supplied by the prior. Depending on the complexity of the underlying statistical model, computing the expression in Eq (5) can be challenging. Through concerted efforts of researchers to develop closed-form solutions and through the explosion of computational power over recent decades that allows applying complex numerical methods (e.g., Monte Carlo sampling and bridge sampling; [56–60]), computing the expression in Eq (5) and variants of it has become feasible. These efforts have led to method developments and software implementations for calculating Bayes factors for survival analyses [61–64] and many other designs (e.g., [42,53,65–71]).

## Choosing priors

Even though it is possible nowadays to calculate Bayes factors for various sorts of designs, it remains difficult to specify an appropriate prior distribution (or prior for short) for the parameters of interest (cf. Eq (5)). The prior is a probability distribution that is placed on the statistical model's parameters of interest and it expresses belief over the plausibility across all possible parameter values before taking into account new data. In the context of null hypothesis Bayes factor calculations, the prior is one important element of the alternative hypothesis. Even among researchers who advocate Bayesian statistics, there is disagreement on how the prior should be specified (see, e.g., [72,73]).

Objective Bayesians strive to define non-informative priors that are as "objective" as possible. Objective Bayesians assert that the results of Bayesian analyses should depend only to a minimal extent on the beliefs of different people. They promote default priors that can be used when no other information is available and often seek to find priors that "behave well" and fulfill certain mathematical properties (see, e.g., [74,75] for more details). Subjective Bayesians, on the other hand, counter that the subjective nature of the prior is an integral part of Bayesian analyses. According to them, the prior allows the incorporation of domain knowledge and results from prior studies into Bayesian analyses and therefore permits tests of theories [32,76]. Further, they state that it is questionable whether a truly "objective" prior even exists.

Recently, the opportunities that well-defined priors open were increasingly recognized in biomedical research. Prior elicitation procedures, in which informed priors are defined by means of using external sources, gained popularity within biomedical research (e.g., [77–81]). There are various forms of prior elicitation. For example, through structured interviews, information about prior beliefs can be extracted by one or multiple experts in the respective field, which is subsequently combined into one prior (see, e.g., [79,80,82,83]). Tools like the MATCH software [84] have been developed for this purpose. Alternatively, the results of meta-analyses and prior research in general can be used to create a prior (e.g., [80,85]). That is, researchers could use the overall effect size combined with a measure of uncertainty from a meta-analysis to construct an informed prior for their own analysis; or they could conduct their own literature search and extract the relevant statistics and use them for developing priors. This approach of using prior study results can also be combined with an empirical Bayes

approach, which utilizes the current data to create a prior instead of predefining it (e.g., [86]). Such a procedure was proposed by [87].

In this article, we follow the approach of using results of prior studies to create priors. For this, we conducted our own literature search instead of relying on meta-analyses. The reason for this is that we aim to suggest priors that are generic, such that they apply to entire medical subfields; most meta-analyses do not offer this generality.

## The current study

In the present article, we develop priors centered around zero for Bayesian Cox regression for nine subfields that we believe are representative for different areas of research within biomedicine. For the construction of these priors we make use of reported hazard ratios and associated $x$% confidence intervals from large corpora of existing studies. These extracted data are then combined through pooling to generate priors. These priors apply to two-sided alternative hypotheses. Since we are ignorant of the direction of effects, our proposed priors can be truncated at 0 for directional hypotheses.

## Methods

We selected the subfields in biomedicine considered for further investigation based on a taxonomy provided by Scimago (available at https://www.scimagojr.com/journalrank.php; [88]). On the Scimago website, we used "Medicine" as a subject area, upon which a list of medical subfields were provided (see Fig 1). Among those, we selected the following nine subfields for further consideration:

1. Anesthesiology and pain medicine
2. Cardiology and cardiovascular medicine
3. Gastroenterology
4. Hematology
5. Immunology and allergy
6. Neurology
7. Oncology
8. Psychiatry and mental health
9. Pulmonary and respiratory medicine

Our selection was based on three criteria: (1) we aimed to obtain a manageable number of subfields (between eight and twelve); (2) we aimed to obtain subfields with limited

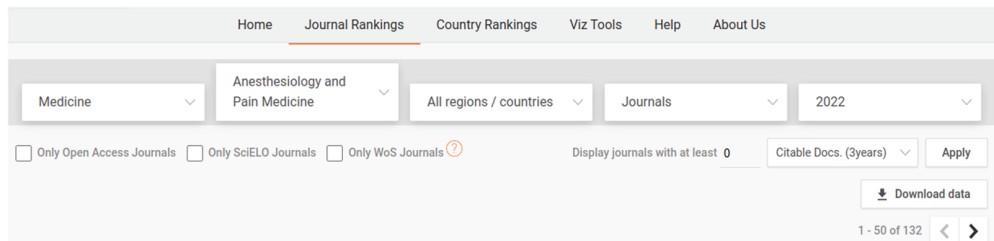

**Fig 1. Settings for extracting top journals from Scimago (available at https://www.scimagojr.com/journalrank.php; [88]) for one of the nine considered subfields in biomedicine; in this case "Anesthesiology and pain medicine".**

overlap; and (3) we aimed to obtain subfields that represent relatively large areas of study within biomedicine. For each of the nine subfields, we obtained a list of the top journals from Scimago. We considered only journals (i.e., neither book series, nor conferences and proceedings, nor trade journals); we considered journals from all regions/countries; and the journal list was based on the year 2022 (see Fig 1 for an example of the settings on the Scimago website for the subfield of "Anesthesiology and pain medicine"). The extraction of the top journals for all nine subfields yielded 2,469 journals in total (see columns 1 and 2 of Table 1). Some subfields shared a set of journals; for example, the journal "Pain" is a top journal for both the subfields of "Anesthesiology and pain medicine" and "Neurology". We found that there was not a lot of overlap of journals between the subfields, with 2,196 of the 2,469 journals being uniquely assigned to only one subfield.

As a separate step, we used Scopus to obtain a list of medical articles. We used the following search query:

```
ABS(("hazard ratio" OR hr) AND cox) AND
SUBJAREA(medi) AND
PUBYEAR > 1999 AND PUBYEAR < 2021 AND
(LIMIT-TO(SRCTYPE,"j")) AND
(LIMIT-TO(PUBSTAGE,"final")) AND
(LIMIT-TO(DOCTYPE,"ar")) AND
(LIMIT-TO(LANGUAGE,"English"))
```

Only fully published (line 5) articles (line 6) from a journal (line 4) written in English (line 7) were considered. Furthermore, the results had to belong to the field of medicine (line 2) and be published between the years 2000 and 2020, inclusive (line 3). Lastly, the abstracts of the results needed to contain the keywords "hazard ratio" or "HR" and the keyword "Cox", ignoring case (line 1). Note, however, that this search query was generic such that it did not restrict the results towards one of the nine subfields. The Scopus query yielded 59,669 results, of which 23 could not be exported, leaving 59,646 results in total (see column 3 of Table 1).

The 59,646 Scopus results were allocated to the nine subfields by matching the journal names indicated by Scopus to the Scimago lists of journal names for the nine subfields.

**Table 1. Number of used journals and studies for each of the nine subfields within biomedicine.**

| Subfield | N journals | N studies allocated | N studies matched | N studies considered |
|---|---|---|---|---|
| Anesthesiology and pain medicine | 44/132 | 360 | 215 | 211 |
| Cardiology and cardiovascular medicine | 230/366 | 10,718 | 6,555 | 6,504 |
| Gastroenterology | 88/157 | 2,088 | 1,311 | 1,300 |
| Hematology | 66/134 | 1,601 | 857 | 849 |
| Immunology and allergy | 82/214 | 1,360 | 839 | 833 |
| Neurology | 178/387 | 2,840 | 1,651 | 1,640 |
| Oncology | 239/373 | 13,163 | 7,741 | 7,684 |
| Psychiatry and mental health | 159/560 | 1,750 | 1,038 | 1,029 |
| Pulmonary and respiratory medicine | 84/146 | 2,551 | 1,561 | 1,548 |
| All | 1,170/2,469 | 36,431/59,646 | 21,768 | 21,598 |

The first column indicates the subfield, the second column the number of used (i.e., matched between Scopus data and Scimago data) journals (not necessarily unique) from all Scimago journals, the third column the number of studies allocated, the fourth column the number of studies for which there was a match and data extraction was successful, and the fifth column the number of studies remaining after excluding studies that provide flawed results.

Importantly, a Scopus result could be allocated to multiple subfields as some subfields had journals in common. Before the allocation of Scopus results to the nine subfields, both the Scopus journal names and the Scimago lists of journal names were cleaned and standardized in order to accommodate slight differences in their presentation. This included replacing "&" and "&" with "AND", removing all characters that are not alphabetic or white space, repositioning the word "the" (e.g., "Lancet Oncology, The" was turned into "The Lancet Oncology"), and transforming all characters to uppercase. The number of matched journal names relative to the total number of Scimago journal names for the nine subfields are shown in column 2 of Table 1 and the number of allocated results for each of the nine subfields can be seen in column 3 of Table 1.

Once the individual Scopus results were allocated to the nine subfields, we extracted hazard ratios and associated $x$% confidence intervals from the abstracts of the results. This was done in an automatic fashion through the use of regular expressions (see [89] for the standard reference on regular expressions). We extracted the following information from the abstracts:

- Hazard ratio (HR)
- Confidence level of the confidence interval (CI) for HR (i.e., $100\,(1-\alpha)$)
- Lower boundary of the CI for HR ($\mathrm{HR}^l$)
- Upper boundary of the CI for HR ($\mathrm{HR}^u$)

There are several important details about our implemented text-mining procedure. First, we exclusively considered the abstracts for data extraction. In an attempt to ascertain that this did not result in bias in the extracted effect sizes, we randomly sampled 100 articles for which we could match results in the abstract with our regular expression and another 100 articles for which we could not match results in the abstract with our regular expression. For these articles, we applied our regular expression on the main text (i.e., not the abstracts). The effect size distributions from this sample looked very similar, indicating no strong evidence for a bias as a result of focusing our search on abstracts only (see Fig 5 in Appendix A). Second, if the regular expression yielded multiple matches for a given abstract, only the first match was considered; any other matches were discarded. The justification for this decision was that we assumed that the main findings are commonly reported first, followed by secondary or exploratory findings.

Third, data extraction was only done when all of the four above-mentioned information were available in the abstract. We disregarded abstracts where results were not complete or were presented in any other form. For instance, presentations of a HR coupled with a $p$-value and potentially a test statistic were ignored. Although this seems like an overly drastic measure, the number of matches was still very high (see column 4 of Table 1). Fourth, we did not distinguish between variations of Cox regression (e.g., multivariate, stratified, multiple predictors).

Fifth, we allowed various forms of the displayed results. For example, the following variations were all captured by our regular expression: "HR = 2.3" (with or without spaces around =), "hazard ratio = 2.3", "hazards ratio = 2.3", "hazard ratios = 2.3", "hazard ratio (HR) = 2.3", "hazards ratio [HR] = 2.3", "HR : 2.3", "H.R. : 2.3", and many more. Thus, we attempted to make the regular expression as flexible as possible, so that it could capture the maximum amount of valid text, while still maintaining a healthy level of restrictions. For more details, please consult our code, available at https://osf.io/ua4ys/. In total, we were able to extract data for 21,768 out of 36,431 results (see column 4 of Table 1).

We applied additional checks on the extracted data to make sure that both the regular expression worked properly and the information in the abstracts themselves was correct. As a first step, we checked whether the confidence level of the CI was between 0 and 100. Second, we tested whether HR, $HR^l$, and $HR^u$ were higher than 0 (because the possible range goes from 0 to $\infty$). Third, we examined whether the log HR was approximately (because of rounding) in the middle of $HR^l$ and $HR^u$. Here, we also excluded results where the log HR and at least one of $HR^l$ or $HR^u$ had the same value due to rounding. Any extracted data that did not fulfill all of these criteria was discarded. Column 5 of Table 1 shows that for all nine subfields only a small number of extracted data had to be excluded (170 out of 21,768 in total), leaving a final number of considered results of 21,598.

With this step completed, the nine subfields and their hazard ratios and associated $x$% confidence intervals were considered separately. For each study $i$ in $i \in \{1, 2, \ldots, N\}$ within one of the nine subfields (where $N$ is the number of studies within one of the nine subfields), the extracted $HR_i$ was log-transformed ($b_i$). Also, the standard error of $b_i$ was calculated based on the confidence interval (e.g., [90]) of HR (i.e., $HR_i^l$ and $HR_i^u$):

$$SE(b_i) = \frac{\log HR_i^u - \log HR_i^l}{2z_i^*}, \tag{6}$$

where $z_i^* = Q(1 - \alpha_i/2)$ and $Q(.)$ is the quantile function of the standard Normal distribution. The sign of $b_i$, however, is meaningless because it depends on how the independent variable is coded. For instance, commonly the control condition is coded with 0 and the experimental condition with 1; occasionally, the opposite is the case, which would reverse the sign of $b_i$. We take this arbitrariness into account in any further calculations.

The obtained $b_i$ and $SE(b_i)$ within a subfield could then be used for the construction of a prior for each subfield separately. We decided to use the Normal distribution for the prior. To obtain a prior, we combined the data through pooling (e.g., [90]). Using this procedure, $b_i$ and $SE(b_i)$ were treated as coming from separate samples that were combined into a single pooled sample. One desirable feature of this pooling method is that $b_i$ values with higher $SE(b_i)$ are central to (rather than discarded for) the calculation of the pooled standard error. In other words, the $SE(b_i)$ around the $b_i$ values had a direct influence on the calculation of the pooled standard error: All else being equal, the higher the $SE(b_i)$ of a sample, the more it would increase the pooled standard error. We deemed this behavior desirable since we wanted the prior to reflect uncertainty. The resulting mean and standard error of a single pooled sample served as the mean and standard deviation of the prior.

We decided against using the inverse-variance weighting procedure that is commonly used in meta-analyses (see, e.g., [91]). The reason for this was that the prior would get increasingly narrow as the corpus of considered studies increases. In addition, $b_i$ with high $SE(b_i)$ values have a relatively low influence (they are less diagnostic in determining the mean), which was undesirable for our purposes of trying to estimate the spread of the prior. Also, we decided to not use the partly empirical Bayes procedure described in [87] because the nature of it prescribes that the current data (i.e., not only the corpus of prior studies) is used to determine the prior.

Assuming a constant $n$ across studies $i$ and taking into consideration the arbitrariness of the sign of $b_i$ by using both $b_i$ and $-b_i$ for each study $i$, we can compute the mean $\mu^p$ of the prior (cf. [90]). Since we consider both $b_i$ and $-b_i$ for each study $i$, $\mu^p$ must necessarily be 0:

$$\mu^p = \frac{n \sum_{i=1}^N b_i + n \sum_{i=1}^N -b_i}{2Nn} = \frac{\sum_{i=1}^N b_i - b_i}{2N} = 0. \tag{7}$$

Similarly, we can compute the standard deviation of the prior $\sigma^p$:

$$\sigma^p = \sqrt{\frac{(n-1)\sum_{i=1}^{N} SE(b_i)^2 + n\sum_{i=1}^{N} b_i^2}{Nn-1}}. \tag{8}$$

In essence, Eq (8) sums up $SE(b_i)^2$ and it sums up $b_i^2$. Thus, all else being equal, $\sigma^p$ increases as any of the two quantities increases. Note that our extracted information did not provide information on the sample size $n_i$ within each study. We used an arbitrary sample size of $n = 200$ for all $n_i$. However, in order to verify the robustness of the assumption of equal sample size per study, we randomly varied $n_i$ across studies, so that it could take on values sampled from $U(10, 10000)$. We repeated this procedure 100,000 times to accommodate many possible arrangements of $n_i$.

## Results

The results for the construction of the Normal priors for the nine subfields through the pooling method [90] can be seen in Fig 2. The panels represent the nine subfields. Histograms show the distributions of $b_i$ (and $-b_i$) for the different corpora, independent of $SE(b_i)$. The red curves show the priors resulting from the application of the pooling method [90]. Descriptive statistics can be found in Appendix B.

For all nine subfields, the center of the Normal prior was located at $\mu^p = 0$ as a necessary consequence of our decision to ignore the sign of $b_i$. The more interesting parameter of the Normal prior is the standard deviation $\sigma^p$ because it reflects both the effect sizes and the uncertainties around them based on past studies within a subfield. From the histograms and the Normal priors it can be seen that the effect sizes, and therefore also the Normal priors, were similar across the nine subfields. $\sigma^p$ ranged between 0.915 for "Psychiatry and mental health" and 1.079 for "Gastroenterology".

Since the calculation of $\sigma^p$ through the pooling method [90] was based on an arbitrary choice of $n = 200$ that was the same for all studies within a corpus of a specific subfield, we also investigated the dependence of $\sigma^p$ on $n$. Fig 3 shows this dependence, where the panels represent the different subfields. The box plots represent the variation of $\sigma^p$ as a function of $n_i$, where $n_i$ for each study $i$ was sampled from $U(10, 10000)$ (i.e., different sample sizes were possible across studies), over 100,000 repetitions.

When $n_i$ were allowed to vary across studies, some variation in $\sigma^p$ was observed, as shown by the box plots. Still, the variation in $\sigma^p$ was not radical enough to question our heuristic of using $n = 200$ for all studies within a corpus of a subfield.

### Example application

To illustrate how the obtained priors can be used, we conducted a Bayesian Cox regression on a medical data set, which is described in [92]. In this study, the authors aimed to examine a drug called Remdesivir, which promised to be beneficial for treating patients with Covid-19. The study included $n = 1,062$ participants, of which $n_c = 521$ were randomly placed in the placebo condition and $n_e = 541$ in the Remdesivir condition. All participants had a Covid-19 infection. The primary analysis focused on the amount of time until participants recovered.

It can be argued that this study falls both in the realm of "immunology and allergy" as well as "pulmonary and respiratory medicine" (i.e., $\sigma^p = 1.023$ and $\sigma^p = 0.999$, respectively). Therefore, we conducted our analyses with both priors. Furthermore, we think that it is always good practice to conduct sensitivity analyses. So, we further extended our used priors to entail

## Histograms of b with corresponding N(0, σᵖ) priors

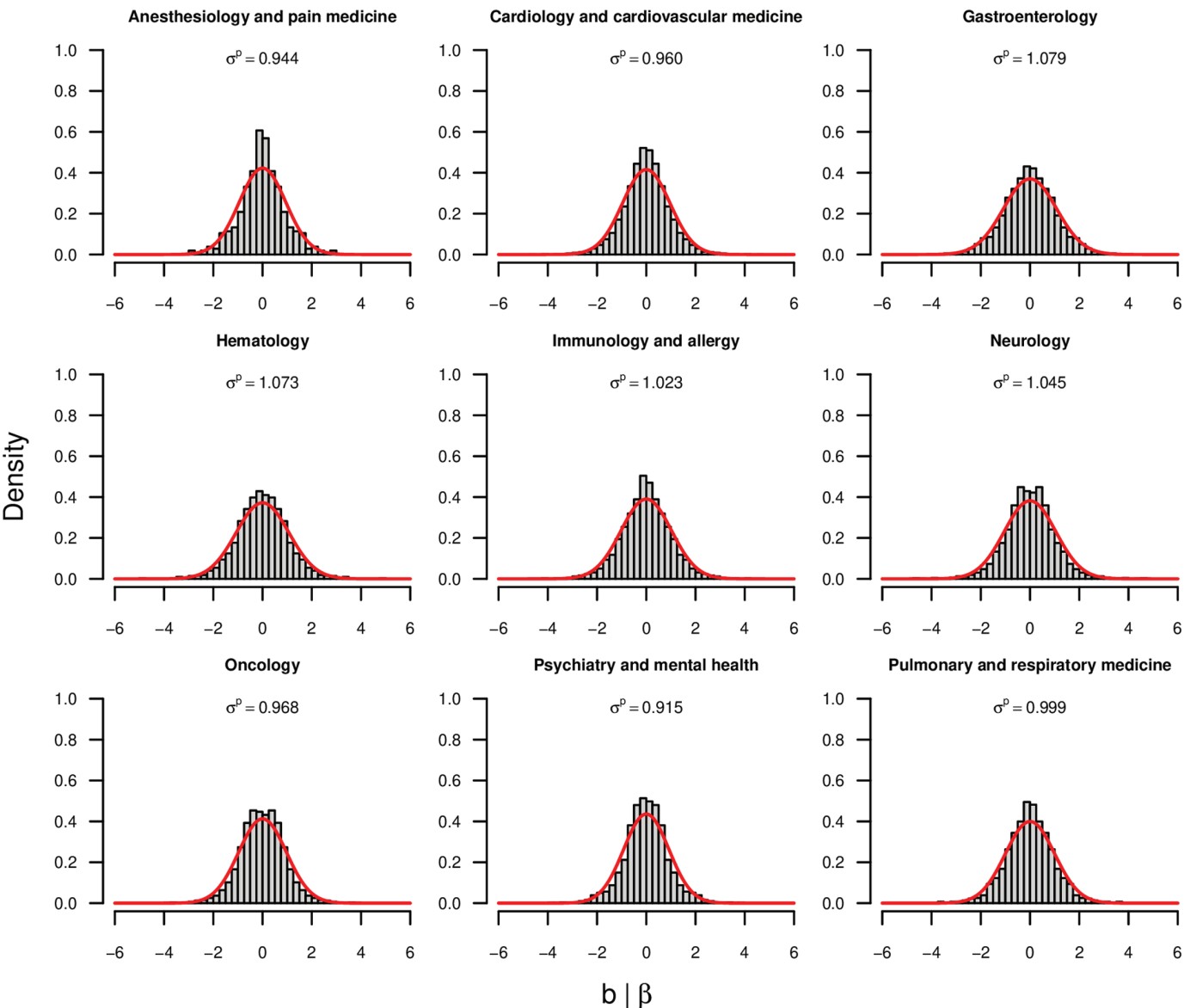

**Fig 2. Prior distributions for the nine subfields, one subfield per panel.** The histograms show the distribution of $b_i$ (and $-b_i$), ignoring $SE(b_i)$. The red curves display the densities of the Normal priors on $\beta$, where $\sigma^p$ is presented at the top of each panel.

Normal priors centered on $\mu^p = 0$ with $\sigma^p \in \{0.3, 0.4, ..., 1.7\}$. We argued that a major advantage of Bayesian statistics is to monitor and optionally stop data collection based on interim results. We demonstrate this procedure on the basis of the data set by [92]. Unfortunately, the data file does not record the order in which participants entered the study. However, for the purposes of this demonstration, we deemed it appropriate to simply take a random order. In essence, we pretend that the study is still ongoing and we stop data collection once the resulting Bayes factor falls outside the decision threshold of 1/20 or 20. In general, the choice

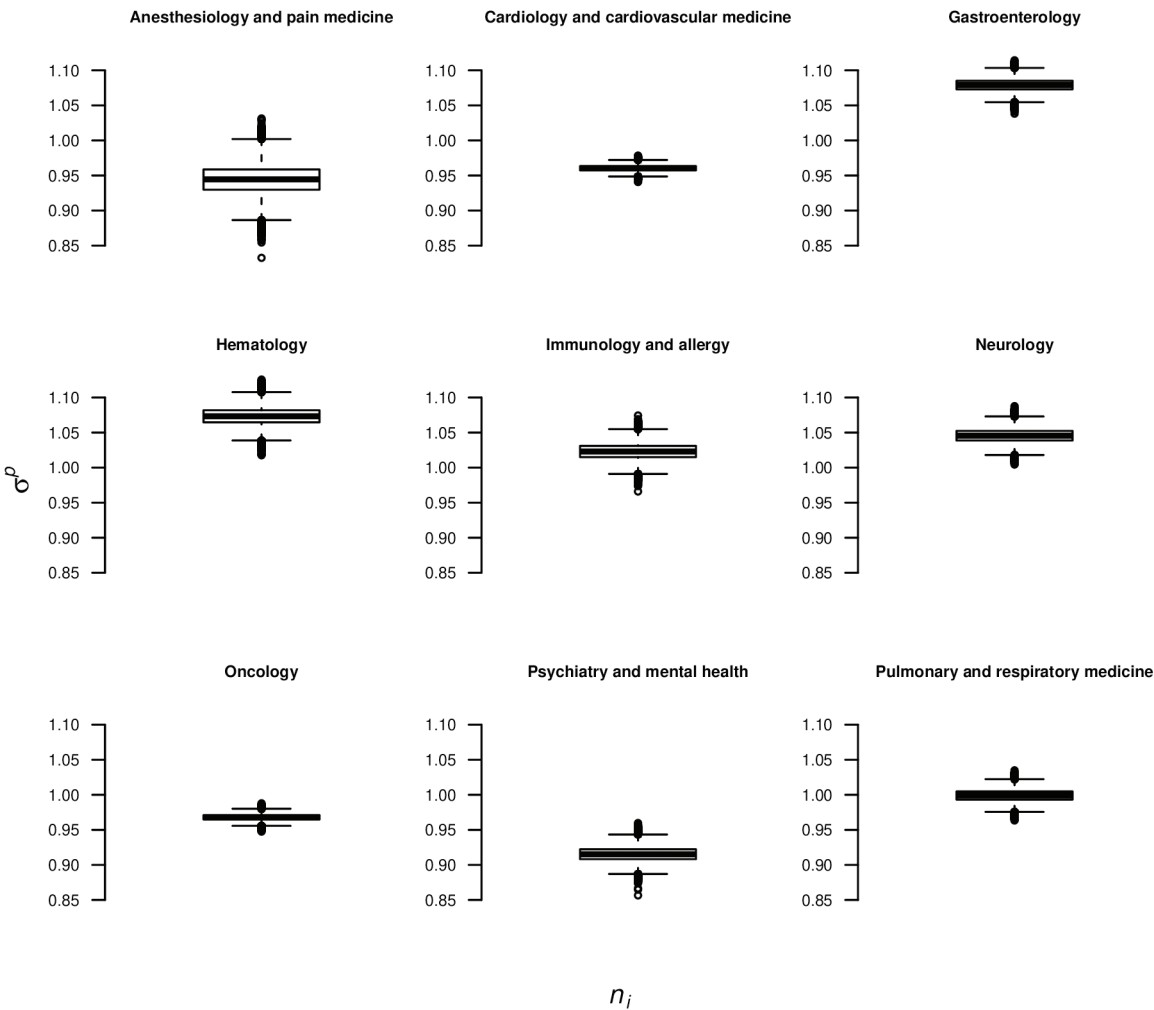

**Fig 3. Sensitivity of the pooling method with respect to the choice of $n_i$ for the estimation of $\sigma^p$.** The nine panels correspond to the nine subfields. The box plots show $\sigma^p$ of the priors when each $n_i$ is drawn randomly from $U(10, 10000)$ (i.e., studies have different sample sizes); this process was repeated 100,000 times.

for a specific decision threshold should be tailored to the research at hand. There are scenarios where the priority lies with more certainty and scenarios where the priority lies with more efficiency. However, researchers should be aware of the tradeoff between certainty and resource demands; the more certain we want to be, the more cases must be sampled. Lastly, for our analysis we use a one-sided positive alternative hypothesis because it is hypothesized that patients receiving Remdesivir recover quicker than patients receiving a placebo.

Fig 4 shows the development of the Bayes factors as new data enter the study for various priors. The blue and red lines correspond to the "immunology and allergy" and "pulmonary and respiratory medicine" priors and the gray lines to the priors used in the sensitivity analyses. It can be seen that all Bayes factor trajectories cross the upper threshold of 20 at some point, demonstrating the robustness of the findings. For some extreme priors, the upper threshold is already crossed at about 300 observations.

**Development of Bayes factors using different priors**

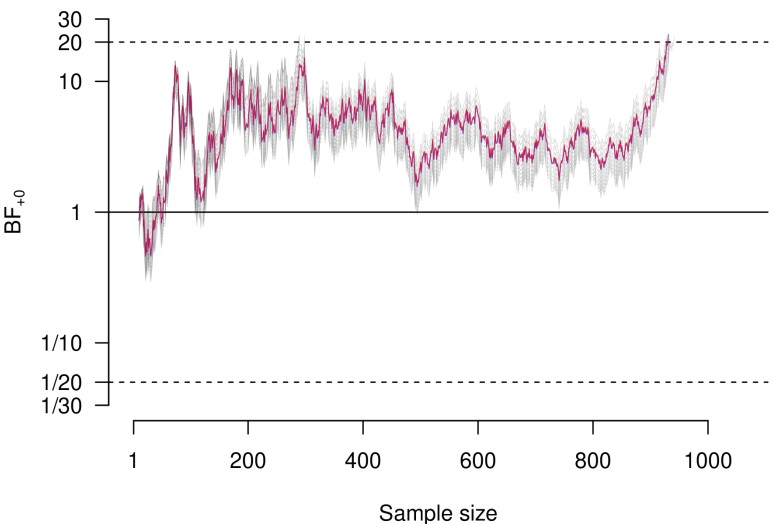

**Fig 4. Results of a Bayesian Cox regression applied to the data set provided by [92].** The blue and red lines show the development of the Bayes factor as new data enter the study for the "immunology and allergy" and "pulmonary and respiratory medicine" priors. The blue and red lines heavily overlap, so the blue line is hardly visible. The gray lines show the development of the Bayes factor for the sensitivity analysis (i.e., various Normal priors). Data collection is stopped once the Bayes factor exceeds 1/20 or 20.

## Discussion

Survival analysis, and in particular Cox regression [38], is an indispensable statistical tool for biomedical research. The ubiquitous frequentist framework is limited in that it cannot quantify evidence in favor of the null hypothesis that there is no effect [42] and in that it does not allow continuing or stopping data collection based on interim analyses [47–50,54,55]. Bayes factors remedy these shortcomings and permit intuitive interpretations. Nevertheless, the specification of priors that are required for Bayesian analyses can be difficult.

In this paper, we have developed priors for the $\beta$ parameter in Cox regression for nine subfields in biomedicine (see column 1 in Table 1). These priors were informed by large corpora of already existing studies within the respective subfields and therefore provide reasonable approximations to the to-be-expected effect sizes and uncertainties thereof. For all nine subfields, we decided to use a Normal prior, which is centered on $\mu^p = 0$. We found very similar standard deviations for the Normal priors across the nine subfields, ranging from $\sigma^p = 0.915$ for "Psychiatry and mental health" to $\sigma^p = 1.079$ for "Gastroenterology", suggesting considerable similarities across subfields. Since our developed priors differ only slightly across the nine subfields, we believe that it is reasonable to use a standard Normal prior (i.e., N (0, 1)) for all nine subfields, forming a compromise among the nine individual priors, as a starting point. Still, any choice of prior is always arbitrary to some degree. Therefore, we urge researchers to complement their analysis using a specific prior with sensitivity analyses (e.g., [31,93–95]), in which parameters of the prior are systematically varied and even entirely different (reasonable) priors are chosen, in order to examine the robustness of the resulting Bayes factors.

We caution the reader to not take our proposed priors to be set in stone. The choice of prior always depends to some extent on the goals of the researcher. For example, it might not always be desirable for the prior to accurately reflect expected effects. Sometimes, the focus

might be on ensuring sufficient shrinkage of the parameter estimate (e.g., [87,96,97]). Furthermore, we strongly advice researchers to only apply these priors in settings where they are appropriate. That is, these priors should only be applied in Cox regressions. We discourage the transfer of these priors to other methods, such as meta analysis or linear regression. In addition, our developed priors represent expected effect sizes of entire medical fields at a very global level. It is likely that researchers often have access to information that is more tailored to their research question. In that case, we encourage researchers to utilize this more informed and problem-specific information for the construction of a prior. We also want to highlight that researchers should define their prior, including the priors for any sensitivity analyses, before they inspect the data. Trying out multiple priors and then reporting those few that yield favorable results, is bad scientific practice.

Moreover, our process of arriving at the priors contained decisions, assumptions, and heuristics that might be questioned. First, the allocation of the articles to the nine subfields based alone on the journal is a drastic heuristic. A proportion of the articles is therefore probably classified into the wrong subfield. Second, the regular expression that we created to extract hazard ratios and associated $x$% confidence intervals from the abstracts of the articles might have been biased to some extent. Some journals have very specific reporting guidelines for the abstracts, which might not have been captured by our regular expression. Thus, it is possible that certain journals were systematically underrepresented in our results. We investigated this potential bias by examining the number of articles that remain after conducting the processing steps shown in Table 1 for each journal in each field. We did not find convincing evidence for a systematic underrepresentation of certain journals (see Appendix C). Third, we only matched abstracts where a hazard ratio is combined with a confidence interval but not abstracts where a hazard ratio is combined with a $p$-value. We made this decision because $p$-values often do not map directly onto the confidence intervals. Additional information on how the $p$-value was calculated would be needed, which is rarely available in abstracts. Fourth, for the Cox regression analyses, we did not differentiate between different types of predictors (e.g., categorical, continuous) and types of analyses (e.g., stratified, multivariate), leaving open the possibility that certain nuances are ignored by our calculations. Fifth, our results are predicated on articles that have come up in the Scopus search engine. Any articles that were published in predatory journals, but that were somehow not screened out as such (see e.g., https://www.elsevier.com/connect/the-guardians-of-scopus), may have introduced bias in the estimate of effect size variance, because of lack of proper peer review. Sixth, we assumed that HR refers to hazard ratio in the abstracts. However, HR could also be an abbreviation for other terms like heart rate. To mitigate this concern, we manually screened a random sample of 88 articles for which we could obtain a match and found that in all cases HR corresponded to hazard ratio.

It is clear that our proposed priors for Bayesian Cox regressions are very generic, such that one individual prior accommodates an entire subfield (or even all nine subfields if we are willing to accept the standard Normal prior). These priors might still be appropriate approximations for smaller specializations within subfields. However, in these cases it might be worthwhile to obtain more informed and precise priors that are tailored to these smaller subfields.

## Conclusion

The analysis of time-to-event data with Cox regression [38] is pervasive in biomedical research. Cox regression combined with Bayes factors has much to offer over traditional frequentist inference because it allows researchers to directly contrast the evidence for the null

and alternative hypotheses [42] and because it allows monitoring results during data collection and continue or stop at any time [47–50,54,55]. These characteristics of Bayes factors have the potential to reduce the waste of scarce resources in biomedical research and especially clinical trials [53,98]. However, the specification of priors for these Bayesian analyses can be challenging and be perceived as overly subjective. We propose default priors in Cox regression that are informed by large corpora of already existing studies for nine subfields. These priors are all Normal distributions centered on 0 with standard deviations that are close to 1. They can be used as a default or starting point for medical researchers and can be augmented with sensitivity analyses.

## Appendix A: Comparison of abstract and main text matches

To mitigate the possibility that the effect size distributions are biased, we took our database of articles found on Scopus that matched any of the journals listed in Scimago, yielding 36,431 articles (see Table 1 in the manuscript). Of those articles, we randomly sampled 100 articles for which we could not match results in the abstract with our regular expression (no-match group); and we randomly sampled 100 articles for which we could match

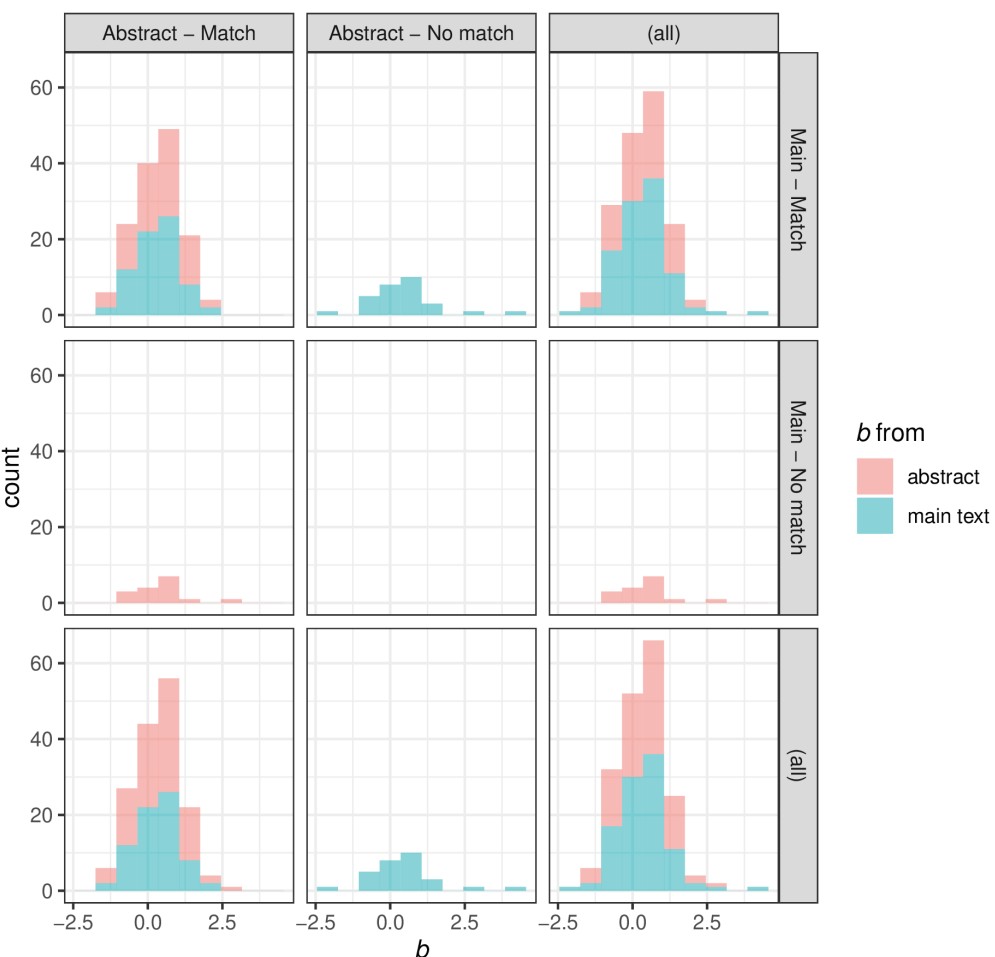

**Fig 5. Effect size distributions for effect sizes matched in the abstract or main text of a sample of 200 articles.**

results in the abstract with our regular expression (match group). Then, we tried to obtain the full texts of these 200 articles. We were able to obtain 86 full texts for the no-match group and 88 full texts for the match group (some articles were not accessible to us). We transformed the PDF files into txt files using the pdftotext command from the Xpdfreader tool https://www.xpdfreader.com/. Then we manually removed the abstracts from the txt files. Subsequently, we used the same regular expression to match results in the main text of the txt files. Fig 5 shows a plot that compares the effect size distributions for extracted results from the abstracts and the main text. As can be seen, the effect size distributions look similar across the different panels, providing evidence that there is probably no strong bias.

## 1 Descriptive statistics for medical fields

See Tables 2, 3, 4, 5, 6, 7, 8, 9, and 10.

## 2 Article retention rate across processing steps

We further investigated the possibility that certain journals are underrepresented in our results by examining the article retention rates over processing steps across fields and journals. In particular, looking at Table 1, it can be seen that there are three processing steps involved: First, Scopus articles are selected based on whether they match the Scimago journal names. Second, articles are selected for which we could obtain a match for the regular expression. Third, of the matched articles, we filtered out articles for which our regular expression yielded ambiguous/wrong results. Fig 6 shows the proportion of articles remaining in a specific field (panels) and journal (lines) when transitioning from the first to the second step (S1) and from the second to the third step (S2). However, journals for which the number of articles is 0 at some processing step are filtered out because the division with 0 is problematic. It can be seen that there is some variation between journals at S1 for all fields. This variation decreases drastically at S2 for all fields.

**Table 2. Descriptive statistics for Anesthesiology and pain medicine.**

|         | $b$    | $SE(b)$ | $b^2$ | $SE(b)^2$ |
|---------|--------|---------|-------|-----------|
| Min     | -2.120 | 0.002   | 0.000 | 0.000     |
| 1st Qu. | -0.010 | 0.069   | 0.043 | 0.005     |
| Median  | 0.336  | 0.164   | 0.261 | 0.027     |
| Mean    | 0.388  | 0.219   | 0.804 | 0.089     |
| 3rd Qu. | 0.782  | 0.308   | 0.781 | 0.095     |
| Max     | 2.896  | 1.211   | 8.386 | 1.468     |

**Table 3. Descriptive statistics for Cardiology and cardiovascular medicine.**

|         | $b$    | $SE(b)$ | $b^2$  | $SE(b)^2$ |
|---------|--------|---------|--------|-----------|
| Min     | -4.247 | 0.000   | 0.000  | 0.000     |
| 1st Qu. | -0.041 | 0.086   | 0.061  | 0.007     |
| Median  | 0.365  | 0.184   | 0.278  | 0.034     |
| y Mean  | 0.407  | 0.238   | 0.824  | 0.099     |
| 3rd Qu. | 0.854  | 0.332   | 0.935  | 0.110     |
| Max     | 4.800  | 1.883   | 23.039 | 3.547     |

**Table 4. Descriptive statistics for Gastroenterology.**

|  | $b$ | $SE(b)$ | $b^2$ | $SE(b)^2$ |
|---|---|---|---|---|
| Min | -3.507 | 0.000 | 0.000 | 0.000 |
| 1st Qu. | -0.174 | 0.110 | 0.090 | 0.012 |
| Median | 0.389 | 0.219 | 0.426 | 0.048 |
| Mean | 0.393 | 0.285 | 1.027 | 0.137 |
| 3rd Qu. | 0.968 | 0.389 | 1.264 | 0.152 |
| Max | 3.459 | 1.274 | 12.296 | 1.623 |

**Table 5. Descriptive statistics for Hematology.**

|  | $b$ | $SE(b)$ | $b^2$ | $SE(b)^2$ |
|---|---|---|---|---|
| Min | -3.219 | 0.001 | 0.000 | 0.000 |
| 1st Qu. | -0.174 | 0.119 | 0.083 | 0.014 |
| Median | 0.378 | 0.225 | 0.380 | 0.051 |
| Mean | 0.398 | 0.275 | 1.029 | 0.123 |
| 3rd Qu. | 0.916 | 0.377 | 1.148 | 0.142 |
| Max | 4.806 | 1.332 | 23.102 | 1.773 |

**Table 6. Descriptive statistics for Immunology and allergy.**

|  | $b$ | $SE(b)$ | $b^2$ | $SE(b)^2$ |
|---|---|---|---|---|
| Min | -2.996 | 0.001 | 0.000 | 0.000 |
| 1st Qu. | -0.128 | 0.105 | 0.069 | 0.011 |
| Median | 0.351 | 0.207 | 0.345 | 0.043 |
| Mean | 0.361 | 0.259 | 0.937 | 0.110 |
| 3rd Qu. | 0.880 | 0.365 | 1.156 | 0.133 |
| Max | 4.248 | 1.145 | 18.050 | 1.311 |

**Table 7. Descriptive statistics for Neurology.**

|  | $b$ | $SE(b)$ | $b^2$ | $SE(b)^2$ |
|---|---|---|---|---|
| Min | -4.605 | 0.001 | 0.000 | 0.000 |
| 1st Qu. | 0.010 | 0.101 | 0.090 | 0.010 |
| Median | 0.425 | 0.202 | 0.339 | 0.041 |
| Mean | 0.446 | 0.255 | 0.978 | 0.115 |
| 3rd Qu. | 0.916 | 0.348 | 1.060 | 0.121 |
| Max | 4.726 | 1.799 | 22.331 | 3.236 |

**Table 8. Descriptive statistics for Oncology.**

|  | $b$ | $SE(b)$ | $b^2$ | $SE(b)^2$ |
|---|---|---|---|---|
| Min | -5.809 | 0.001 | 0.000 | 0.000 |
| 1st Qu. | -0.288 | 0.106 | 0.081 | 0.011 |
| Median | 0.285 | 0.206 | 0.316 | 0.042 |
| Mean | 0.263 | 0.252 | 0.830 | 0.107 |
| 3rd Qu. | 0.765 | 0.338 | 0.856 | 0.115 |
| Max | 5.955 | 2.784 | 35.466 | 7.749 |

**Table 9. Descriptive statistics for Psychiatry and mental health.**

|  | $b$ | $SE(b)$ | $b^2$ | $SE(b)^2$ |
|---|---|---|---|---|
| Min | -2.659 | 0.001 | 0.000 | 0.000 |
| 1st Qu. | 0.068 | 0.069 | 0.062 | 0.005 |
| Median | 0.392 | 0.151 | 0.261 | 0.023 |
| Mean | 0.454 | 0.194 | 0.768 | 0.070 |
| 3rd Qu. | 0.815 | 0.259 | 0.840 | 0.067 |
| Max | 3.886 | 1.428 | 15.099 | 2.040 |

**Table 10. Descriptive statistics for Pulmonary and respiratory medicine.**

|          | $b$     | $SE(b)$ | $b^2$  | $SE(b)^2$ |
|----------|---------|---------|--------|-----------|
| Min      | -3.507  | 0.000   | 0.000  | 0.000     |
| 1st Qu.  | -0.136  | 0.093   | 0.068  | 0.009     |
| Median   | 0.351   | 0.203   | 0.336  | 0.041     |
| Mean     | 0.378   | 0.255   | 0.886  | 0.112     |
| 3rd Qu.  | 0.884   | 0.357   | 1.012  | 0.127     |
| Max      | 3.728   | 1.499   | 13.899 | 2.246     |

# Article retention rate per journal across processing steps

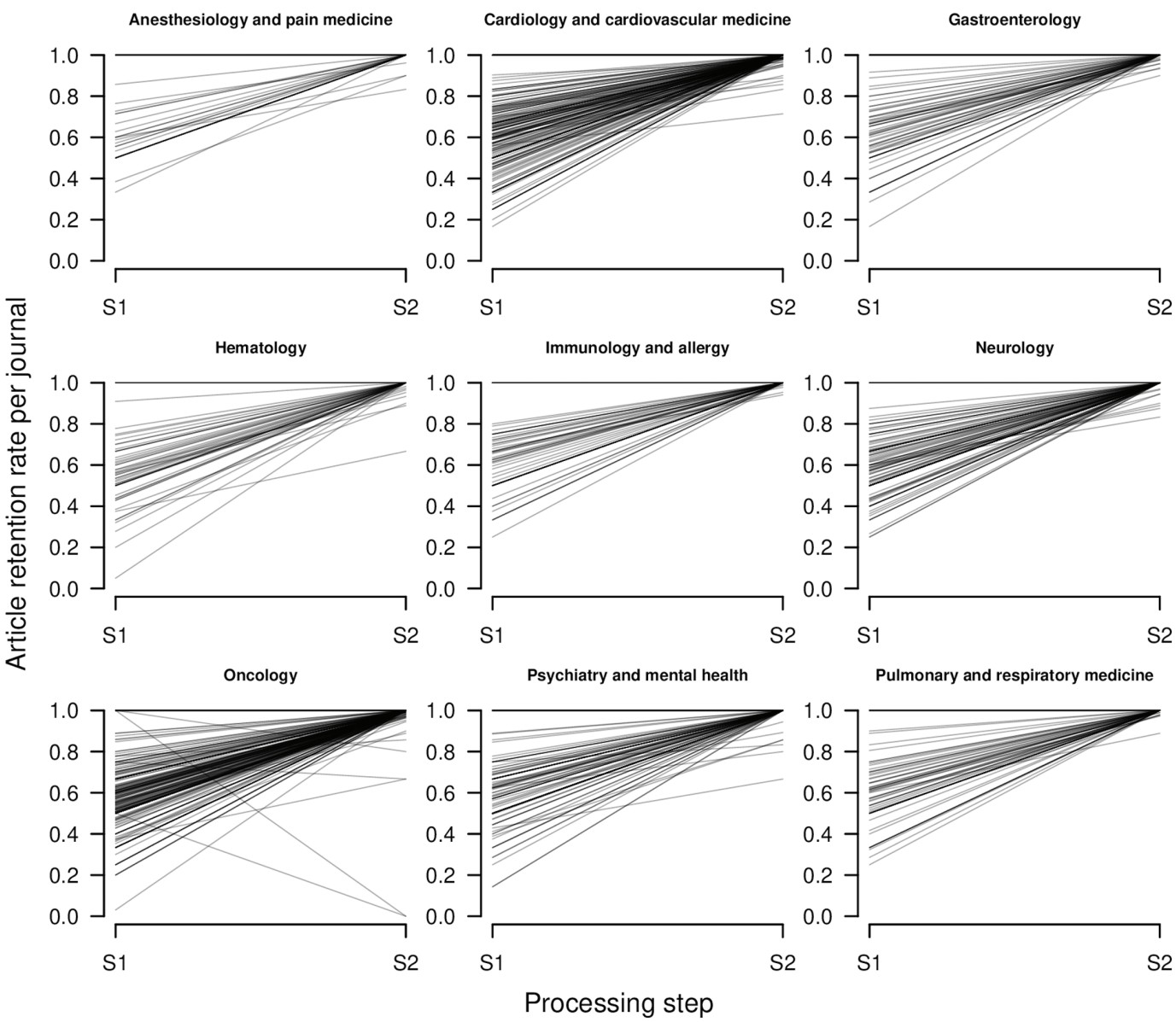

**Fig 6. Proportion of articles remaining in a specific field (panels) and journal (lines) when transitioning from the first to the second step (S1) and from the second to the third step (S2).** Note that journals for which the number of articles is 0 at some processing step were not considered here.

## Author contributions

**Conceptualization:** Maximilian Linde, Laura Jochim, Jorge N. Tendeiro, Don van Raven-zwaaij.

**Formal analysis:** Maximilian Linde.

**Funding acquisition:** Jorge N. Tendeiro, Don van Ravenzwaaij.

**Methodology:** Jorge N. Tendeiro, Don van Ravenzwaaij.

**Software:** Maximilian Linde.

**Supervision:** Jorge N. Tendeiro, Don van Ravenzwaaij.

**Visualization:** Maximilian Linde.

**Writing – original draft:** Maximilian Linde.

**Writing – review & editing:** Laura Jochim, Jorge N. Tendeiro, Don van Ravenzwaaij.

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
