## [Decision Letter · Decision Letter 0]

7 Aug 2024

PONE-D-24-18152Data-driven prior elicitation for Bayes factors in Cox regression for nine subfields in biomedicinePLOS ONE

Dear Dr. Linde,

Thank you for submitting your manuscript to PLOS ONE. After careful consideration, we feel that it has merit but does not fully meet PLOS ONE’s publication criteria as it currently stands. Therefore, we invite you to submit a revised version of the manuscript that addresses the points raised during the review process.

We look forward to receiving your revised manuscript.

Kind regards,

Robin Haunschild

Academic Editor

PLOS ONE

2. In your Methods section, please include additional information about your dataset and ensure that you have included a statement specifying whether the collection and analysis method complied with the terms and conditions for the source of the data. 

Additional Editor Comments (if provided):

Reviewers' comments:

Reviewer's Responses to Questions

**Comments to the Author**

1. Is the manuscript technically sound, and do the data support the conclusions?

Reviewer #1: Yes

Reviewer #2: Yes

2. Has the statistical analysis been performed appropriately and rigorously? 

Reviewer #1: N/A

Reviewer #2: Yes

3. Have the authors made all data underlying the findings in their manuscript fully available?

Reviewer #1: Yes

Reviewer #2: Yes

4. Is the manuscript presented in an intelligible fashion and written in standard English?

Reviewer #1: Yes

Reviewer #2: Yes

5. Review Comments to the Author

Reviewer #1: Summary:

The manuscript proposes “informed default“ prior distributions for the beta coefficient in Cox regression in nine subfields of biomedical research. The priors are based on the analysis of Cox regression coefficients in a large corpus of biomedical studies.

Review:

Finding suitable prior distributions in Bayesian modeling is challenging, and the present manuscript proposes a reasonable way to address this challenge. The methodology for prior specification followed in the manuscript has been applied in similar ways before, but to my knowledge, not for Cox regressions in biomedical science. Given the popularity of survival models and the increasing uptake of Bayesian statistics across scientific fields, I believe that the manuscript can make a valuable contribution to the literature. However, I also have several concerns about the paper, particularly about the validity of the data extraction procedure and the appropriateness of the application example and the simulation study. I will go into more detail on my concerns below and I recommend that the authors address these concerns in a revised version of the manuscript.

I appreciate that the authors used such a large corpus of studies as a basis for prior specification. However, I am concerned that the way the regression coefficients were extracted from the literature may introduce bias into the results. Specifically, estimates were extracted from articles that mentioned the method as well as estimates and confidence intervals in the abstract. This makes me wonder: How often does it happen at all that the specific statistical method is mentioned in the abstract (rather than only in the results section of the article)? To what extent do articles that mention the method (and specific estimates) in the abstract differ systematically from articles that do not mention these details in the abstract? It seems like the extraction method may give preference to articles with structured abstracts (in my view, the most minor bias since the abstract structure is often guided by journal policies rather than by article content), but potentially also to articles where a statistician/methodologist was involved (hence the stronger focus on statistical methods and outcomes), or to articles where a significant result or large effect size was found (something “worth mentioning” in an abstract). It would be good if the authors could run some robustness analyses to check the unbiasedness of extracted effect sizes (e.g., compare full-text search for Cox regression in some journal issues to abstract-only search and compare the results to get a feeling for the potential bias).

Related to my previous point, I think that there are a few details about the extraction method that could be made clearer in the manuscript. Specifically, the authors mention that they extracted articles from “top journals” in the respective fields, but the definition of a “top journal” remained vague. How did the authors ensure that no predatory (pseudo-peer reviewed) journals were part of the selection? Moreover, when explaining the search algorithm using regular expressions, the examples include different versions of “HR”. How can we be sure that “HR” always refers to a hazard ratio and isn’t an abbreviation of something else (e.g., heart rate) and that the reported numbers hence refer to a different quantity? And finally, the extraction method for the standard error assumes two-sided confidence intervals. Intuitively, it makes sense that this would be the standard, but wouldn’t it be possible that some articles report one-sided confidence intervals (e.g., with a one-sided test of treatment effectiveness?).

For me, the application example presented on pp. 9-12 does not fall into the intended application domain of the specified prior distributions. The prior distribution was elicited for the beta coefficient in Cox regressions, but in the presented application example, the prior distribution was applied to the fixed effect in a meta-analysis across five Cox regressions. I would argue that the specified prior distribution is not a good match for the analysis presented in the application example because the meta-analytic effect size estimate tends to be less extreme than the effect size estimate in a single study. Specifically, we can expect that due to sampling variability, the effect size in a single (potentially small) study is more extreme than the combined estimate across multiple studies (weighed by the study sample size). An appropriate prior specification procedure for meta-analytic effect size estimates in Cox regressions that is similar to the described procedure would therefore, in my view, not specify the prior based on a corpus of individual studies, but on a corpus of meta-analyses. I would therefore highly recommend to exchange the example application in the paper with a Bayesian Cox regression in a single study.

In addition to my previous point, I believe that the presented application example falls short of showcasing the advantages of the methodology and the recommended best practices for using the informed prior distribution. The authors repeatedly mention in the introduction of the manuscript that two advantages of the Bayesian statistical framework are the possibility of quantifying evidence for the null hypothesis and the possibility of flexible sequential designs. I believe that it would be a great opportunity to showcase these advantages in the application example by using a sequential hypothesis test with the specified priors and quantifying the evidence for H0 and H1. Moreover, the discussion section of the manuscript mentions that sensitivity analyses for prior distributions are considered best practice, yet the application example does not include sensitivity analyses. I think that here, too, it would be a good opportunity to showcase this best practice in the application example using the specified prior distributions of different (adjacent?) subfields as well as a default prior distribution.

If my calculations are right, the simulation study presented on p. 11 / Figure 3 is unnecessary. The authors decide to assume the same sample size for all studies. This means that Equation 10 can be drastically simplified and basically yields an analytic result for the simulations. Specifically, the index j is unnecessary to describe the sample size n (n is a constant), since all studies are assumed to have the same sample size. Therefore, the terms under the square root can be simplified to:

∑(n-1) θ_j^2=(n-1)∑θ_j^2

∑n(ω_j-μ^p )=n∑(ω_j-0)^2=n∑ω_j^2

∑_(i=1)^K▒n_j -1=nK-1

With this simpler form of the equation, it also becomes relatively clear that the “mirroring” process in the manuscript has no effect on the results. We can simply write the equation in terms of original effect sizes. For each study, the following is true (note that SE(beta) = SE(-beta)):

(n-1)∑θ_j^2=(n-1)(〖SE〗_j^2+〖SE〗_j^2 )=2〖SE〗_j^2 (n-1)

n∑ω_j^2=n(β_j^2+(-β_j )^2 )=2nβ_j^2

This means, we can express Equation 10 as:

√((2(n-1)∑SE_j^2+2n∑β_j^2)/(n⋅2N-1))=√(((n-1)∑SE_j^2+n∑β_j^2)/(nN-1))

From this, we can see that the influence of the sample size of a single study, n, influences the comparative weighting of SE and beta in the equation: In the most extreme cases, beta gets weighed twice as much as SE (for n = 2) or beta and SE obtain equal weights (for n=Inf), respectively. We can simply plug in the assumed sample sizes (10:10000), as well as the sum of SEs and sum of betas into the equation to analyze the sensitivity of results.

In my view, a more sensible way to test the sensitivity of results to the sample sizes would be to draw sets of different sample sizes from realistic distributions (I believe that there are papers on sample size / statistical power development across time in the biomedical sciences?) and use these to determine sigma^p. This way, n would actually differ per study and different studies would be weighed differently in each iteration of the simulation. To make it more realistic, the SEs of individual studies should probably also be negatively correlated with the sample size in the simulation (higher sample size, lower SE).

In the discussion section, the authors write that “it is possible that certain journals were systematically underrepresented in our results”. To me, this sounds like an empirical question that could be answered using the extracted data? If the proportion of articles selected from each journal within a field does not decrease at the same rate throughout the selection process (e.g. the step indicated in Table 1), then there is some bias. This could for example be represented in some sort of Sankey diagram.

In the introduction, I think it could be made clearer that there is a rich tradition of sequential designs in frequentist statistics, too. In particular, in my knowledge, clinical biomedical studies are one of the key application domains of Group Sequential designs. I agree with the authors that Bayesian methods are more flexible, but I think this point may need a bit of a more nuanced discussion.

Also in the introduction, it could be made clearer that the specification of prior distributions is not only necessary in the context of Bayesian hypothesis testing with Bayes factors, but also in the context of Bayesian parameter estimation. I understand that the focus of the article is on Bayes factors, but to readers with little experience in Bayesian statistics, this might lead to misunderstandings.

Generally, I believe that in practice “informed default” priors are unfortunately often applied outside of their intended application domain (e.g., as has happened to the Oosterwijk prior that is applied outside of social psychology and wasn’t even initially intended as a default). Of course, how the priors will be used in practice is outside of the control of the authors. However, I believe that a thorough discussion of the limitations of recommended use in the discussion section of the manuscript could be beneficial to restrict their use to reasonable applications. For example, I would argue that the specified priors are not ideal for meta-analytic effect size parameters (see my point above) and that they should not be used outside of Cox regression (this may be obvious to the authors, but maybe not to non-experts in Bayesian analyses). Moreover, if more field-specific result summaries exist (e.g., Cochrane meta-analysis on breast cancer treatments), and a new analysis is run in this particular field (e.g., new breast cancer treatment), then it might be good to create a more informed prior distribution based on these field-specific results.

Another aspect that I believe warrants further discussion is how the mix of results for the null- and alternative hypothesis in the literature influences the specified prior distributions. Specifically, the specified prior distributions are intended for use under the alternative hypothesis. However, unless there is severe publication bias, it can be assumed that the literature contains effect sizes that are generated under the null- and alternative hypothesis. From a theoretical standpoint, I am wondering: Should effect sizes that were generated under the null hypothesis really be included to specify a prior for the alternative hypothesis?

Related to my previous point, I believe that the topic of how publication bias might have influenced the specified prior distributions warrants further discussion. Ideally, it would be great to see some p-curves or z-curves together to contextualize the results, but since p-values were not extracted, I understand that this is difficult.

In the results section, it would be great if the authors could include some descriptive statistics of the betas and SEs in the different subfields, and perhaps also the sum of beta^2 and the sum of SE^2 for each discipline. Given the extent to which the results depend on these statistics (see also my earlier points), it would be great to have some more information available in the manuscript.

Reading the manuscript, I was wondering a little what prior distribution the authors would recommend for directional tests. For example, if I assume that a new treatment is better than an old treatment, I would want to test a directional test on the beta parameter. Should I just use the specified distribution for my field and truncate it at zero (e.g., a truncated Normal(0, 0.96) for cardiology)? Or would it be better to use a non-central informed prior distribution? What could be the advantages/disadvantages of each?

Signed

Angelika Stefan

Reviewer #2: Motivated by the problem of facilitating Bayesian inference of Cox regression, the authors proposed a data-driven prior elicitation method. Overall, the presentation is clear and the proposed method is sensible. I believe this work can be further strengthened by more numerical results and a more thorough discussion on how we can prevent Bayes factor hacking in practice.

Numerical Results:

- Can the authors run some simulation studies to quantify the benefits of using these data-driven priors? For example, I imagine that the data-driven priors would carry information from prior studies, and hence would reduce the sample size needed to reach similar strength of evidence compared to another study with weak informative prior.

- Can the authors add a sensitivity analysis in the example application? I believe such an analysis would help better demonstrate the usefulness of the proposed method and set a good example for practitioners to follow.

Discussion on how we can prevent Bayes factor hacking:

- In frequentist statistics, there is a notion of p-value hacking (https://en.wikipedia.org/wiki/Data_dredging). Following similar logic, it is natural to think that could be Bayes factor hacking in the Bayesian world. Therefore, I believe it would be important for authors to give a more thorough discussion on how we can prevent Bayes factor hacking when the prior distribution can be manipulated in the name of data-driven prior.

6. PLOS authors have the option to publish the peer review history of their article (what does this mean?). If published, this will include your full peer review and any attached files.

Reviewer #1: **Yes: **Angelika M. Stefan

Reviewer #2: No

---

## [Author Response · Author response to Decision Letter 1]

6 Dec 2024

Dear Dr. Haunschild,

Thank you for considering our manuscript “Data-driven prior elicitation for Bayes factors in Cox regression for nine subfields in biomedicine” for publication in PLOS ONE. We have read your and the reviewers’ comments carefully. Below you find our responses. The reviewers’ comments are shown in regular font and our responses are shown in bold font. Our revision contains a version of our manuscript without track changes (i.e., a clean version with all changes already incorporated) and a version of our manuscript with track changes. For the latter, track changes that correspond to additions are shown in blue font and track changes that correspond to removals are shown in red font.

We look forward to your comments.

Kind regards,

Maximilian Linde

Laura Jochim

Jorge N. Tendeiro

Don van Ravenzwaaij 

Dear Dr. Linde,

Thank you for submitting your manuscript to PLOS ONE. After careful consideration, we feel that it has merit but does not fully meet PLOS ONE’s publication criteria as it currently stands. Therefore, we invite you to submit a revised version of the manuscript that addresses the points raised during the review process.

● A rebuttal letter that responds to each point raised by the academic editor and reviewer(s). You should upload this letter as a separate file labeled 'Response to Reviewers'.

● A marked-up copy of your manuscript that highlights changes made to the original version. You should upload this as a separate file labeled 'Revised Manuscript with Track Changes'.

● An unmarked version of your revised paper without tracked changes. You should upload this as a separate file labeled 'Manuscript'.

We look forward to receiving your revised manuscript.

Kind regards,

Robin Haunschild

Academic Editor

PLOS ONE

2. In your Methods section, please include additional information about your dataset and ensure that you have included a statement specifying whether the collection and analysis method complied with the terms and conditions for the source of the data.

Additional Editor Comments (if provided):

E1-1: We are grateful for the opportunity to revise and resubmit our manuscript.

Reviewer #1: Summary:

The manuscript proposes „informed default“ prior distributions for the beta coefficient in Cox regression in nine subfields of biomedical research. The priors are based on the analysis of Cox regression coefficients in a large corpus of biomedical studies.

Review:

Finding suitable prior distributions in Bayesian modeling is challenging, and the present manuscript proposes a reasonable way to address this challenge. The methodology for prior specification followed in the manuscript has been applied in similar ways before, but to my knowledge, not for Cox regressions in biomedical science. Given the popularity of survival models and the increasing uptake of Bayesian statistics across scientific fields, I believe that the manuscript can make a valuable contribution to the literature. However, I also have several concerns about the paper, particularly about the validity of the data extraction procedure and the appropriateness of the application example and the simulation study. I will go into more detail on my concerns below and I recommend that the authors address these concerns in a revised version of the manuscript.

R1-1: We thank the reviewer for the positive evaluation and the review. We address the reviewer’s comments below.

I appreciate that the authors used such a large corpus of studies as a basis for prior specification. However, I am concerned that the way the regression coefficients were extracted from the literature may introduce bias into the results. Specifically, estimates were extracted from articles that mentioned the method as well as estimates and confidence intervals in the abstract. This makes me wonder: How often does it happen at all that the specific statistical method is mentioned in the abstract (rather than only in the results section of the article)? To what extent do articles that mention the method (and specific estimates) in the abstract differ systematically from articles that do not mention these details in the abstract? It seems like the extraction method may give preference to articles with structured abstracts (in my view, the most minor bias since the abstract structure is often guided by journal policies rather than by article content), but potentially also to articles where a statistician/methodologist was involved (hence the stronger focus on statistical methods and outcomes), or to articles where a significant result or large effect size was found (something “worth mentioning” in an abstract). It would be good if the authors could run some robustness analyses to check the unbiasedness of extracted effect sizes (e.g., compare full-text search for Cox regression in some journal issues to abstract-only search and compare the results to get a feeling for the potential bias).

R1-2: We understand the concerns of the reviewer. Therefore, we have done additional investigations regarding this potential bias. Specifically, we took our database of articles found on Scopus that matched any of the journals listed in Scimago, yielding 36,431 articles (see Table 1 in the manuscript). Of those articles, we randomly sampled 100 articles for which we could not match results in the abstract with our regular expression (no-match group); and we randomly sampled 100 articles for which we could match results in the abstract with our regular expression (match group). Then, we tried to obtain the full texts of these 200 articles. We were able to obtain 86 full texts for the no-match group and 88 full texts for the match group (some articles were not accessible to us). We transformed the PDF files into txt files using the pdftotext command from the Xpdfreader tool (https://www.xpdfreader.com/). The resulting txt files were not perfect but they should suffice for the current demonstration. Then we manually removed the abstracts from the txt files. Subsequently, we used the same regular expression as in our original submission to match results in the main text of the txt files. Below is a plot that compares the effect size distributions for extracted results from the abstracts (the original findings) and the main text. As can be seen, the effect size distributions look similar across the different panels, providing evidence that there is probably no strong bias. We have added the following to an Appendix: “To mitigate the possibility that the effect size distributions are biased, we took our database of articles found on Scopus that matched any of the journals listed in Scimago, yielding 36,431 articles (see Table 1 in the manuscript). Of those articles, we randomly sampled 100 articles for which we could not match results in the abstract with our regular expression (no-match group); and we randomly sampled 100 articles for which we could match results in the abstract with our regular expression (match group). Then, we tried to obtain the full texts of these 200 articles. We were able to obtain 86 full texts for the no-match group and 88 full texts for the match group (some articles were not accessible to us). We transformed the PDF files into txt files using the pdftotext command from the Xpdfreader tool (https://www.xpdfreader.com/). Then we manually removed the abstracts from the txt files. Subsequently, we used the same regular expression to match results in the main text of the txt files. Fig 5 shows a plot that compares the effect size distributions for extracted results from the abstracts and the main text. As can be seen, the effect size distributions look similar across the different panels, providing evidence that there is probably no strong bias.”

Related to my previous point, I think that there are a few details about the extraction method that could be made clearer in the manuscript. Specifically, the authors mention that they extracted articles from “top journals” in the respective fields, but the definition of a “top journal” remained vague. How did the authors ensure that no predatory (pseudo-peer reviewed) journals were part of the selection? Moreover, when explaining the search algorithm using regular expressions, the examples include different versions of “HR”. How can we be sure that “HR” always refers to a hazard ratio and isn’t an abbreviation of something else (e.g., heart rate) and that the reported numbers hence refer to a different quantity? And finally, the extraction method for the standard error assumes two-sided confidence intervals. Intuitively, it makes sense that this would be the standard, but wouldn’t it be possible that some articles report one-sided confidence intervals (e.g., with a one-sided test of treatment effectiveness?).

R1-3: In the following, we address the three points raised by the reviewer:

1) We think that it is almost impossible to be certain that all articles contained in our corpus are not connected to predatory journals. There are several databases that list predatory journals, the most famous one being Beall’s list (https://beallslist.net/). However, these databases are often outdated or incomplete. Due to these drawbacks, we have decided to not filter out articles based on these lists. An alternative would have been to check the journals in our corpus manually. But with 1170 considered journals, this task would have exceeded our resources. Even more, we believe that this alternative would not be worth the effort because it would be prone to errors; false positives and false negatives would probably arise because of the uncertainties surrounding classification of journals to be predatory or non-predatory. Ultimately, we decided to not screen for predatory journals for two reasons. First, there is a workforce installed at Scopus that constantly screens for predatory journals in their database (https://www.elsevier.com/connect/the-guardians-of-scopus). Second, even if a small proportion of the journals in our corpus can be considered predatory, we think that their influence on our results is minimal, given the large number of non-predatory journals. To accommodate the point of the reviewer, we have added the following limitation to our Discussion section: “Fifth, our results are predicated on articles that have come up in the Scopus search engine. Any articles that were published in predatory journals, but that were somehow not screened out as such (see e.g., https://www.elsevier.com/connect/the-guardians-of-scopus), may have introduced bias in the estimate of effect size variance, because of lack of proper peer review.”

2) We cannot be absolutely certain that “HR” in the regular expression matches hazard ratios only. We agree that “HR” could in principle also refer to other abbreviations, such as heart rate. However, our regular expression does not only match “HR”. Instead, it also requires that “HR” is immediately followed by some form of confidence interval. This requirement drastically decreases the probability that the regular expression captures heart rate or anything else than hazard ratio. Further, all our abstracts must contain “Cox”, further decreasing the chances that HR refers to something other than hazard ratio. To mitigate this concern even more, we manually screened the 88 articles mentioned in R1-2 and added the following to the Discussion section: “Sixth, we assumed that HR refers to hazard ratio in the abstracts. However, HR could also be an abbreviation for other terms like heart rate. To mitigate this concern, we manually screened a random sample of 88 articles for which we could obtain a match and found that in all cases HR corresponded to hazard ratio.”

3) The way that our regular expression is constructed increases the probability that only articles that report two-sided confidence intervals are matched. More precisely, our regular expression has a match when both the lower and upper limits of a confidence interval are represented by a number, which is higher than 0. Importantly, this excludes 0, which would be required for upper one-sided confidence intervals. Moreover, any symbols representing infinity are also excluded, which would be required for lower one-sided confidence intervals. In addition, our regular expression method checks whether point estimates of the hazard ratio are approximately in the middle of the reported confidence interval (see paragraph ‘We applied … results of 21,598’). Note that only a small proportion of confidence intervals was excluded as a result of this check (see Table 1), strongly suggesting that the included confidence intervals are all two-sided.

For me, the application example presented on pp. 9-12 does not fall into the intended application domain of the specified prior distributions. The prior distribution was elicited for the beta coefficient in Cox regressions, but in the presented application example, the prior distribution was applied to the fixed effect in a meta-analysis across five Cox regressions. I would argue that the specified prior distribution is not a good match for the analysis presented in the application example because the meta-analytic effect size estimate tends to be less extreme than the effect size estimate in a single study. Specifically, we can expect that due to sampling variability, the effect size in a single (potentially small) study is more extreme than the combined estimate across multiple studies (weighed by the study sample size). An appropriate prior specification procedure for meta-analytic effect size estimates in Cox regressions that is similar to the described procedure would therefore, in my view, not specify the prior based on a corpus of individual studies, but on a corpus of meta-analyses. I would therefore highly recommend to exchange the example application in the paper with a Bayesian Cox regression in a single study.

In addition to my previous point, I believe that the presented application example falls short of showcasing the advantages of the methodology and the recommended best practices for using the informed prior distribution. The authors repeatedly mention in the introduction of the manuscript that two advantages of the Bayesian statistical framework are the possibility of quantifying evidence for the null hypothesis and the possibility of flexible sequential designs. I believe that it would be a great opportunity to showcase these advantages in the application example by using a sequential hypothesis test with the s

---

## [Decision Letter · Decision Letter 1]

15 Jan 2025

PONE-D-24-18152R1Data-driven prior elicitation for Bayes factors in Cox regression for nine subfields in biomedicinePLOS ONE

Dear Dr. Linde,

Thank you for submitting your manuscript to PLOS ONE. After careful consideration, we feel that it has merit but does not fully meet PLOS ONE’s publication criteria as it currently stands. Therefore, we invite you to submit a revised version of the manuscript that addresses the points raised during the review process.

We look forward to receiving your revised manuscript.

Kind regards,

Robin Haunschild

Academic Editor

PLOS ONE

Journal Requirements:

Reviewers' comments:

Reviewer's Responses to Questions

**Comments to the Author**

1. If the authors have adequately addressed your comments raised in a previous round of review and you feel that this manuscript is now acceptable for publication, you may indicate that here to bypass the “Comments to the Author” section, enter your conflict of interest statement in the “Confidential to Editor” section, and submit your "Accept" recommendation.

Reviewer #1: (No Response)

Reviewer #2: All comments have been addressed

2. Is the manuscript technically sound, and do the data support the conclusions?

Reviewer #1: Yes

Reviewer #2: Yes

3. Has the statistical analysis been performed appropriately and rigorously? 

Reviewer #1: Yes

Reviewer #2: Yes

4. Have the authors made all data underlying the findings in their manuscript fully available?

Reviewer #1: Yes

Reviewer #2: No

5. Is the manuscript presented in an intelligible fashion and written in standard English?

Reviewer #1: Yes

Reviewer #2: Yes

6. Review Comments to the Author

Reviewer #1: I think the manuscript is in a very good shape in its current form and the authors have done an excellent job to address the points I raised in my previous review. I particularly want to praise the thorough robustness analyses that have been conducted in response to my earlier comments and the introduction of a new example that much better demonstrates the application of the methodology. I do have a few really minor comments on the current version, particularly on the new example application, but in my view, these should not impede the publication of the manuscript.

• In the application example, the manuscript reads “It can be seen that all Bayes factor trajectories cross the upper threshold of 30 at some point”. However, Figure 4 seems to suggest that the trajectories cross the threshold of 20, but not of 30? Is there potentially an issue with the axis labels? Furthermore, it may be helpful to very briefly discuss the definition of stopping thresholds in the new paragraph about the sequential design (i.e., why is 30 a good choice?)

• In the example application, is there any benefit of using the data-driven priors compared to the default priors? Judging based on Figure 4, it looks a bit like the data-driven priors yield Bayes factors in the upper third of the distribution, which would indicate at least some efficiency benefit over the default priors. I understand that the authors are reluctant to point to the potential efficiency benefits of informed priors (I’ve gathered as much from the response to the second reviewer). However, I think the application example shows that at least IF the parameter in the present study is similar to the corpus of literature used for deriving the prior, then efficiency gains can be expected compared to SOME default priors. Moreover, the variability in data-driven priors is much smaller than in the other priors, potentially indicating greater stability of results if priors are chosen in a principled manner.

• In Figure 4, it is difficult to distinguish the red and blue lines. Is there any way to overcome this issue (perhaps decreasing the line width and/or making the grey lines in the background brighter would help?), or if not, could the authors perhaps briefly note the overlap in the figure caption so that readers don’t get confused?

• In response to my earlier comment R1-7, the authors added a section to the introduction that contains the sentence “A prominent way to deal with this issue within the frequentist framework is the use of sequential designs” – I think this should be group sequential designs?

• I thought the new figure illustrating the article retention rate across processing steps was interesting – will this be included in the appendix / supplementary materials?

Signed,

Angelika Stefan

Reviewer #2: (No Response)

7. PLOS authors have the option to publish the peer review history of their article (what does this mean?). If published, this will include your full peer review and any attached files.

Reviewer #1: **Yes: **Angelika M. Stefan

Reviewer #2: No

---

## [Author Response · Author response to Decision Letter 2]

12 Mar 2025

Dear Dr. Haunschild,

Thank you for considering our manuscript “Data-driven prior elicitation for Bayes factors in Cox regression for nine subfields in biomedicine” for publication in PLOS ONE. We have read your and the reviewers’ comments carefully. Below you find our responses. The reviewers’ comments are shown in regular font and our responses are shown in bold font. Our revision contains a version of our manuscript without track changes (i.e., a clean version with all changes already incorporated) and a version of our manuscript with track changes. For the latter, track changes that correspond to additions are shown in blue font and track changes that correspond to removals are shown in red font.

We look forward to your comments.

Kind regards,

Maximilian Linde

Laura Jochim

Jorge N. Tendeiro

Don van Ravenzwaaij 

PONE-D-24-18152R1

Data-driven prior elicitation for Bayes factors in Cox regression for nine subfields in biomedicine

PLOS ONE

Dear Dr. Linde,

Thank you for submitting your manuscript to PLOS ONE. After careful consideration, we feel that it has merit but does not fully meet PLOS ONE’s publication criteria as it currently stands. Therefore, we invite you to submit a revised version of the manuscript that addresses the points raised during the review process.

We look forward to receiving your revised manuscript.

Kind regards,

Robin Haunschild

Academic Editor

PLOS ONE

Journal Requirements:

E1-1: We thank the editor for the positive evaluation of our manuscript and for the opportunity to resubmit our manuscript. 

Reviewer 1:

I think the manuscript is in a very good shape in its current form and the authors have done an excellent job to address the points I raised in my previous review. I particularly want to praise the thorough robustness analyses that have been conducted in response to my earlier comments and the introduction of a new example that much better demonstrates the application of the methodology. I do have a few really minor comments on the current version, particularly on the new example application, but in my view, these should not impede the publication of the manuscript.

R1-1: We thank the reviewer for the very positive evaluation of our revised manuscript.

In the application example, the manuscript reads “It can be seen that all Bayes factor trajectories cross the upper threshold of 30 at some point”. However, Figure 4 seems to suggest that the trajectories cross the threshold of 20, but not of 30? Is there potentially an issue with the axis labels? Furthermore, it may be helpful to very briefly discuss the definition of stopping thresholds in the new paragraph about the sequential design (i.e., why is 30 a good choice?)

R1-2: We are grateful that the reviewer caught this error. It turned out that the figure is correct but the text is wrong. That is, we used a threshold of 20. We have corrected the text accordingly. Further, we followed the reviewer's request to give some additional explanation about stopping thresholds. We wrote the following: “In general, the choice for a specific decision threshold should be tailored to the research at hand. There are scenarios where more certainty is desired compared to other scenarios. However, researchers should be aware of the tradeoff between certainty and resource demands; the more certain we want to be, the more cases must be sampled.”. We think that a more detailed discussion about this extensive topic is beyond the scope of our article.

In the example application, is there any benefit of using the data-driven priors compared to the default priors? Judging based on Figure 4, it looks a bit like the data-driven priors yield Bayes factors in the upper third of the distribution, which would indicate at least some efficiency benefit over the default priors. I understand that the authors are reluctant to point to the potential efficiency benefits of informed priors (I’ve gathered as much from the response to the second reviewer). However, I think the application example shows that at least IF the parameter in the present study is similar to the corpus of literature used for deriving the prior, then efficiency gains can be expected compared to SOME default priors. Moreover, the variability in data-driven priors is much smaller than in the other priors, potentially indicating greater stability of results if priors are chosen in a principled manner.

R1-3: In general, we are very much in favour of using informed priors when relevant information is available. In our case, however, we did not observe efficiency gains of the data-driven priors. Although based on eyeballing, it appears that the data-driven priors result in Bayes factors in the upper third of the distribution generated by the default priors used in the sensitivity analyses, numerical confirmation revealed that it is closer to the median. In an additional analysis, we looked at potential Bayes factor decision thresholds of {3, 4, …, 60}. For almost all of these Bayes factor decision thresholds, the rank of the data-driven priors, in terms of the speed of reaching that threshold, was in the middle, compared to the default priors we used in our sensitivity analysis. In other words, roughly 50% of the priors reached the decision thresholds faster and 50% slower. We observed that in our scenario, priors that are more concentrated at 0 (i.e., smaller standard deviation) reached the decision thresholds faster.

In Figure 4, it is difficult to distinguish the red and blue lines. Is there any way to overcome this issue (perhaps decreasing the line width and/or making the grey lines in the background brighter would help?), or if not, could the authors perhaps briefly note the overlap in the figure caption so that readers don’t get confused?

R1-4: To mitigate this issue, we have now decreased the line width and increased the transparency of lines. These steps improved the visibility of the lines. However, since the blue and red lines overlap so heavily, the visual distinction is still not perfect. Therefore, we added another note to the caption of Figure 4: “The blue and red lines heavily overlap, so the blue line is hardly visible.”

In response to my earlier comment R1-7, the authors added a section to the introduction that contains the sentence “A prominent way to deal with this issue within the frequentist framework is the use of sequential designs” – I think this should be group sequential designs?

R1-5: We have now corrected this sentence according to the reviewer's request.

I thought the new figure illustrating the article retention rate across processing steps was interesting – will this be included in the appendix / supplementary materials?

R1-6: We have now included and explained the figure in an appendix.

---

## [Decision Letter · Decision Letter 2]

18 Mar 2025

Data-driven prior elicitation for Bayes factors in Cox regression for nine subfields in biomedicine

PONE-D-24-18152R2

Dear Dr. Linde,

We’re pleased to inform you that your manuscript has been judged scientifically suitable for publication and will be formally accepted for publication once it meets all outstanding technical requirements.

Kind regards,

Robin Haunschild

Academic Editor

PLOS ONE

Additional Editor Comments (optional):

Reviewers' comments:

Reviewer's Responses to Questions

**Comments to the Author**

1. If the authors have adequately addressed your comments raised in a previous round of review and you feel that this manuscript is now acceptable for publication, you may indicate that here to bypass the “Comments to the Author” section, enter your conflict of interest statement in the “Confidential to Editor” section, and submit your "Accept" recommendation.

Reviewer #1: All comments have been addressed

2. Is the manuscript technically sound, and do the data support the conclusions?

Reviewer #1: Yes

3. Has the statistical analysis been performed appropriately and rigorously? 

Reviewer #1: Yes

4. Have the authors made all data underlying the findings in their manuscript fully available?

Reviewer #1: Yes

5. Is the manuscript presented in an intelligible fashion and written in standard English?

Reviewer #1: Yes

6. Review Comments to the Author

Reviewer #1: All my comments were addressed. I thank the authors for considering the concerns that I previously raised and for their thoughtful responses. In my view, this manuscript is ready to be published.

7. PLOS authors have the option to publish the peer review history of their article (what does this mean?). If published, this will include your full peer review and any attached files.

Reviewer #1: **Yes: **Angelika M. Stefan

---

## [Editor Report · Acceptance letter]

PONE-D-24-18152R2

PLOS ONE

Dear Dr. Linde,

I'm pleased to inform you that your manuscript has been deemed suitable for publication in PLOS ONE. Congratulations! Your manuscript is now being handed over to our production team.

Kind regards,

on behalf of

Dr. Robin Haunschild

Academic Editor

PLOS ONE